# On Certified Generalization in Structured Prediction

**Bastian Boll**
Image & Pattern Analysis Group
Heidelberg University
bastian.boll@iwr.uni-heidelberg.de

**Christoph Schnörr**
Image & Pattern Analysis Group
Heidelberg University
schnoerr@math.uni-heidelberg.de

## Abstract

In structured prediction, target objects have rich internal structure which does not factorize into independent components and violates common i.i.d. assumptions. This challenge becomes apparent through the exponentially large output space in applications such as image segmentation or scene graph generation. We present a novel PAC-Bayesian risk bound for structured prediction wherein the rate of generalization scales not only with the number of structured examples but also with their size. The underlying assumption, conforming to ongoing research on generative models, is that data are generated by the Knothe-Rosenblatt rearrangement of a factorizing reference measure. This allows to explicitly distill the structure between random output variables into a Wasserstein dependency matrix. Our work makes a preliminary step towards leveraging powerful generative models to establish generalization bounds for discriminative downstream tasks in the challenging setting of structured prediction.

## 1 Introduction

### 1.1 Overview

*Structured prediction* is the task of predicting an output which itself contains internal structure. As an example, consider the problem of image segmentation. The output to be predicted is an assignment of semantic classes to each image pixel. However, the segmentation problem is not merely a pixelwise classification, because each pixel is not independently assigned a semantic class. If two pixels are adjacent in the image plane, it is more likely that they belong to the same class than to different ones. A way to remedy this problem is to enumerate all possible segmentations of an image and treat the problem as classification. However, the output space of this classification is now exponentially large. This exponential (in the number of pixels) size of the output space indicates a much more difficult problem than the unstructured case of classification. In fact, it also presents an immediate challenge for any statistical learning theory which requires assumptions on the number of output classes [44].

From a practical perspective, structured prediction problems present a challenge of label aquisition. For instance, dense manual segmentation of an image is significantly more labour intensive than manual classification. In turn, one would expect that pixelwise segmentation of an image contains much richer information to be exploited in supervised learning. However, this is not represented in typical statistical learning theory which assumes all data to be independently drawn from the true underlying distribution. In the extreme case of only a single structured example being available for training, these statistical learning theories can not make any meaningful statement on generalization.

Addressing this point in particular, [38] presents an analysis of the dependency structure in the output and proves a risk certificate which decays in both the number of structured examples *m and their size d*. Refering back to the example of image segmentation, this amounts to a high-probability bound on the fraction of misslabeled pixels which *decays with the number of labeled pixels* observed during training as opposed to merely the number of segmented images.

At the core of statistical learning theory lies the concentration of measure phenomenon which posits that a stable function of a large number of weakly dependent random variables will take values close to its mean [37, 9]. This is relevant because model risk, the expected loss on unseen data, is the mean of empirical risk under the draw of the sample. Learning theories can thus be built on concentration of measure results. In the present work, we focus on PAC-Bayesian learning theory [51, 43, 42]. For an overview of PAC-Bayesian theory we refer to [13, 27, 1].

In addition to a concentration result such as a moment-generating function (MGF) bound, PAC-Bayesian arguments employ a change of measure, typically via Donsker and Varadhan's variational formula. In particular, stochastic predictors, i.e. distributions over a hypothesis class of models are considered. A core objective is to construct generalization bounds which hold uniformly over all stochastic predictors called *PAC-Bayes posteriors*. Model complexity is then measured as relative entropy to a reference stochastic predictor called *PAC-Bayes prior*. As this terminology alludes to, the PAC-Bayes posterior is informed by more data than the PAC-Bayes prior. Note however, that PAC-Bayesian theory generalizes Bayesian theory because prior and posterior are not typically connected via likelihood.

PAC-Bayesian theory has gained considerable attention in recent years due to the demonstration of non-vacuous *risk certificates in deep learning* [23]. Since then, a line of research has succeeded in tightening the risk bounds of deep classifiers [48, 47, 16]. In addition, multiple authors have studied ways to weaken underlying assumptions of bounded loss [29, 28] and i.i.d. data [3].

The majority of recent works in this field focuses on classification or regression. *Structured* prediction has received comparatively little recent attention, an exception being the work [11] which provides a PAC-Bayesian perspective on the implicit loss embedding framework for structured prediction [15].

## 1.2   Related Work

Here, we continue a line of research started by [39, 40, 38] which aims to construct PAC-Bayesian risk bounds for structured prediction that account for generalization from a single (but large) structured datum. Instrumental to their analysis is the stability of inference and quantified dependence in the data distribution. The latter is expressed in terms of $\vartheta$-mixing coefficients, the total variation distance between data distributions conditioned on fixed values of a subset of variables. For structured prediction with Hamming loss, a coupling of such conditional measures can be constructed [24] such that $\vartheta$-mixing coefficients yield an upper-bound that allows to invoke concentration of measure arguments [36]. The result is an MGF bound which the authors employ in a subsequent PAC-Bayesian construction – achieving generalization from a single datum.

The underlying assumption of these previous works is that data are generated by a Markov random field (MRF). This model assumption is somewhat limiting because Markov properties, certain conditional independences, likely do not hold for many real-world data distributions. In addition, MRFs are difficult to work with computationally. Exact inference in MRFs is NP-hard [57] and thus learning, which often contains inference as a subproblem, presents significant computational roadblocks. Even once an MRF has been found which represents data reasonably well, numerical evaluation of the PAC-Bayesian risk certificate proposed in [38] will again be computationally difficult.

Nevertheless, we share the sentiment of previous authors that some assumption on the data-generating process is required in structured prediction. This is because conditional data distributions, the distribution of data conditioned on a fixed set of values for a subset of variables, are central to establishing concentration of measure via the martingale method [35]. Consider again the example of image segmentation. Once we have fixed a sufficiently large number of pixels to arbitrary values (and class labels), even a large dataset will not contain an abundance of data which match these values and thus provide statistical power to learn the conditional distribution. This problem is well-known in conditional density estimation [56].

## 1.3   Contribution, Organization

In this work, we propose to instead assume a triangular and monotone transport, a *Knothe-Rosenblatt (KR) rearrangement* [32, 49, 12, 8, 41] of a reference measure as data model. This choice is attractive for multiple reasons. First, any data distribution which does not contain atoms can be represented

uniquely in this way [6] which should suffice to represent many distributions of practical interest. With regard to conditional distributions, the KR-rearrangement has the convenient property that conditioning on a fixed value for a subset of variables can again be represented by KR-rearrangement. We will use this property in our construction of coupling measures between conditional distributions.

Specifically,

- We present a novel PAC-Bayesian risk bound for structured prediction wherein the rate of generalization scales not only with the number of structured examples but also with their size.

- Based on data generated by KR-rearrangement of a tractable reference measure, we distill relevant structure of the data distribution into a Wasserstein dependency matrix. Our analysis hinges on state-of-the-art results in concentration theory [35] which serve to bound moment-generating functions by properties of the Wasserstein dependency matrix. We subsequently invoke a PAC-Bayesian argument to derive the desired risk certificate.

- We also propose to leverage a construction of bad input data as a computational tool to find entries of the Wasserstein dependency matrix.

We stress the fact that many established approaches to generative modelling can be seen as instances of measure transport. For instance, it includes normalizing flows [54, 53, 33, 46, 50], diffusion models [52, 30], generative adversarial networks and variational autoencoders [10, 26]. While most measure transport models which currently enjoy empirical success are not KR-rearrangements, we hope that the methods presented here can lay the foundation of leveraging powerful generative models to build risk certificates for discriminative downstream tasks.

**Organization**   The central concepts of concentration theory and measure transport are described in the preliminary Sections 2 and 3. The main results are presented in Section 4. In Section 5 we present a first discussion of computational aspects related to the presented framework. The paper closes on a discussion of limitations in Section 6 and a conclusion in Section 7.

**Basic Notation**   For any $d \in \mathbb{N}$, denote $[d] = \{1, \ldots, d\}$. If $z \in \mathcal{Z}^d$ is a vector, we refer to the subvector of entries with index in a set $I \subseteq [d]$ as $z^I$. In particular, index sets of interest will be half-open and closed intervals $(i, d] \subseteq [d]$ and $[i, d] \subseteq [d]$. Analogously, we will index the output of vector-valued functions $f^I$ and marginal measures $\mu^I$. For a set $\mathcal{B} \subseteq \mathcal{Z}^d$, we denote its complement in $\mathcal{Z}^d$ by $\mathcal{B}^c = \mathcal{Z}^d \setminus \mathcal{B}$ and for a measure $\mu$ on $\mathcal{Z}^d$, we denote the conditional measure given $\mathcal{B}^c$ as $\mu | \mathcal{B}^c$. If $Z$ is a random variable with distribution $\mu$ on $\mathcal{Z}^d$ and $I, J \subseteq [d]$ are disjoint index sets with $I \cup J = [d]$, we denote the conditional law of $Z^I$ given $Z^J = z^J$ as $\mu(dz^I | z^J)$.

## 2   Concentration of Measure and Generalization

Let $\mathcal{X}$ denote an input space and $\mathcal{Y}$ denote an output space. Let $\mu$ be a distribution on $\mathcal{Z}^d = (\mathcal{X} \times \mathcal{Y})^d$. There are two restrictions inherent to this setup. First, an input is always paired with an output and thus the number of inputs needs to match the number of outputs. Second, all structured data will be drawn from $\mu$ and thus the size of each structured datum will be the same. Otherwise, $\mathcal{X}$ and $\mathcal{Y}$ can in principle be arbitrary sets which admit metrics. For concreteness, think of $\mathcal{X} = [0, 1] \subseteq \mathbb{R}$ as being a set of gray values and $\mathcal{Y} = \mathbb{R}$ containing signed distances from a semantic boundary [45] in an image with $d$ pixels. In this case, $\mathcal{Z}^d$ contains all binary segmentations of grayvalue images.

The goal of learning is to find parameters $\theta$ which define a predictor $\phi_\theta \colon \mathcal{X}^d \to \mathcal{Y}^d$ such that the *risk*

$$\mathcal{R}(\theta) = \mathbb{E}_{(X,Y) \sim \mu}[L(\phi_\theta(X), Y)] \tag{1}$$

with respect to some loss function $L \colon \mathcal{Y}^d \times \mathcal{Y}^d \to \mathbb{R}$ is minimized. However, the true risk (1) is typically intractable because $\mu$ is unknown. A related tractable quantity is the *empirical risk*

$$\mathcal{R}_m(\theta, \mathcal{D}_m) = \frac{1}{m} \sum_{k \in [m]} L(\phi_\theta(X^{(k)}), Y^{(k)}) \tag{2}$$

of a sample $\mathcal{D}_m = (X^{(k)}, Y^{(k)})_{k \in [m]}$ drawn from $\mu^m$. We further assume that the loss of structured outputs is the mean of bounded *pointwise* loss $\ell \colon \mathcal{Y} \times \mathcal{Y} \to [0, 1]$

$$L(\widetilde{y}^{(k)}, y^{(k)}) = \frac{1}{d} \sum_{i \in [d]} \ell(\widetilde{y}_i^{(k)}, y_i^{(k)}) \,. \tag{3}$$

In PAC-Bayesian constructions, we consider stochastic predictors $\zeta$, i.e. measures on a hypothesis space $\mathcal{H}$ of predictors $\phi_\theta \colon \mathcal{X}^d \to \mathcal{Y}^d$ as identified with measures on the underlying parameter space from which $\theta$ is selected. We then take the above notions of risk and empirical risk in expectation over parameter draws

$$\mathcal{R}(\zeta) = \mathbb{E}_{\theta \sim \zeta}[\mathcal{R}(\theta)], \qquad \mathcal{R}_m(\zeta, \mathcal{D}_m) = \mathbb{E}_{\theta \sim \zeta}[\mathcal{R}_m(\theta, \mathcal{D}_m)] \,. \tag{4}$$

The expected value of empirical risk with respect to the sample is the true risk. For this reason, a central tool for the study of generalization is the *concentration of measure* phenomenon. Informally, it states that a stable function of many weakly dependent random variables concentrates on its mean. We will invoke a line of reasoning put forward in [35] and propose a novel approach to structured prediction based on the measure-transport framework outlined in Section 1. To this end, we first define the following formal notions of *stability* and *dependence*.

Let $\rho$ be a metric such that $\mathcal{Z}$ has finite diameter

$$\|\rho\| = \sup_{\xi, \xi' \in \mathcal{Z}} \rho(\xi, \xi') < \infty \tag{5}$$

and let $\rho^d(z, z') = \sum_{i \in [d]} \rho(z_i, z_i')$ denote the corresponding product metric on $\mathcal{Z}^d$.

**Definition 1 (Local oscillation).** *Let $f \colon \mathcal{Z}^d \to \mathbb{R}$ be Lipschitz with respect to $\rho^d$. Then the quantities*

$$\delta_i(f) = \sup_{z, z' \in \mathcal{Z}^d, z'_{[d] \setminus \{i\}} = z_{[d] \setminus \{i\}}} \frac{|f(z) - f(z')|}{\rho(z_i, z_i')}, \qquad i \in [d] \tag{6}$$

*are called the local oscillations of $f$.*

The vector of local oscillations gives a granular account of stability. In order to discuss interdependence of data in a probability space $(\mathcal{Z}^d, \mu, \Sigma)$, define the Markov kernels

$$K^{(i)}(z, dw) = \delta_{z^{[i-1]}}(dw^{[i-1]}) \otimes \mu^{[i,d]}(dw^{[i,d]} | z^{[i-1]}), \qquad i \in [d] \tag{7}$$

as well as $K^{(d+1)}(z, dw) = \delta_z(dw)$ and their action on functions

$$K^{(i)} f(z) = \int f(y) K^{(i)}(z, dw) = \int f(z^{[i-1]} w^{[i,d]}) \mu^{[i,d]}(dw^{[i,d]} | z^{[i-1]}) \tag{8}$$

where in the edge case $i = 1$, the condition on $z^{[i-1]} = z^{\{\}}$ is removed. Here $K^{(i)}(z, dw)$ is a Borel measure for every $z$ and (8) computes the expected value of $f$ at $z$, conditioned on the fixed realization of the subvector $z^{[i-1]}$. It turns out that the effect of the kernel (7) on local oscillations serves to quantify dependence of data with joint distribution $\mu$.

**Definition 2 (Wasserstein matrix).** *For $i \in [d+1]$, let $K^{(i)}$ denote the Markov kernel (8). A matrix $V^{(i)} \in \mathbb{R}_{\geq 0}^{d \times d}$ is called a Wasserstein matrix [25] for $K^{(i)}$, if*

$$\delta_k(K^{(i)} f) \leq \sum_{j \in [d]} V_{kj}^{(i)} \delta_j(f), \qquad \forall k \in [d] \tag{9}$$

*for any function $f \colon \mathcal{Z}^d \to \mathbb{R}$ which is Lipschitz with respect to $\rho^d$.*

The two concepts defined above will be used in Section 4 to construct a moment-generating function bound via the martingale method.

## 3 Triangular Measure Transport

Suppose a structured output is composed of $d > 0$ unstructured data in a space $\mathcal{Z}$. Then the target measure $\mu$ of interest is a measure on $\mathcal{Z}^d$ which does not factorize into simpler distributions. A popular method of representing complex joint distributions of interdependent random variables is to define a map $T \colon \mathcal{Z}^d \to \mathcal{Z}^d$ which transports a tractable *factorizing* reference measure $\nu^d$ to the target measure $\mu$, i.e. $T_\sharp \nu^d = \mu$. This abstract framework encompasses many generative models such as normalizing flows [54, 53, 33, 46, 50], diffusion models [52, 30], generative adversarial networks and variational autoencoders [10, 26]. Here, we focus on transport maps $T$ which are monotone and triangular in the sense that $T(z)_i$ only depends on the inputs $z^{[i]}$ and each $T(z^{[i-1]}, \cdot)_i$ is an increasing function. Such a map is called a *Knothe-Rosenblatt (KR) rearrangement* [32, 49, 12, 8, 41]. If both $\nu^d$ and $\mu$ have no atoms then the KR rearrangement exists and is unique [6]. In particular, normal distribution $\nu^d$ and any absolutely continuous (with respect to the Lebesgue measure) distribution $\mu$ meet these criteria. The KR rearrangement has the useful property that certain conditional distributions have a simple representation.

**Lemma 3 (Lemma 1 of [41]).** *Let $T \colon \mathcal{Z}^d \to \mathcal{Z}^d$ be the KR-rearrangement which satisfies $T_\sharp \nu^d = \mu$. For arbitrary $i \in [d]$, let $z^{[i]} \in \mathcal{Z}^i$ be fixed. Then*

$$\mu(dw^{(i,d]}|z^{[i]}) = T^{(i,d]}(\overline{z}^{[i]}, \cdot)_\sharp \nu^{d-i} \tag{10}$$

*where $\overline{z}^{[i]}$ is the unique element of $\mathcal{Z}^i$ such that $T^{[i]}(\overline{z}^{[i]}) = z^{[i]}$.*

Numerical realization of KR rearrangements has recently received attention [5] and more broadly, a variety of triangular transport architectures exists [20, 21]. However, we do not focus on numerical considerations in the present theoretical work.

One may wonder if data generated by KR-rearrangement implicitly restricts the choice of possible input and output spaces since the monotonicity requirement on $T$ can only be satisfied if the underlying set $\mathcal{Z}$ is ordered. However, many feature spaces of practical interest are still permissible for what follows. In particular, both $\mathcal{X}$ and $\mathcal{Y}$ may be Euclidean spaces or hypercubes. Inkeeping with the introductory image segmentation example, suppose $\mathcal{X} = [0,1]^3$ contains RGB color values and $\mathcal{Y} = [-1,1]$ contains signed distance from a semantic boundary in the image plane. Then we can interpret $\mathcal{Z}^d = ([0,1]^3 \times [-1,1])^d$ as a product of compact intervals in $\mathbb{R}^{4d}$ which is clearly permissible as underlying space for KR-rearrangement. All presented results hold irrespective of whether $\mathcal{Z}$ contains such internal dimensions as long as a natural ordering of the set $\mathcal{Z}$ is induced.

## 4 PAC-Bayesian Risk Certificate

In this section we present a novel PAC-Bayesian risk bound for structured prediction which combines three main ingredients.

(1) A concentration of measure theorem for dependent data (Theorem 4) which builds on the notion of a Wasserstein dependency matrix;

(2) a simple construction of coupling measures between conditional distributions (Lemma 5) which serves to represent the Wasserstein dependency matrix;

(3) a PAC-Bayesian argument (Theorem 7) employing Donsker-Varadhan's variational formula in concert with concentration of measure results.

The first theorem summarizes key results from [35] on the concentration of measure phenomenon for dependent random variables. We have slightly generalized by augmenting the underlying Doob martingale construction with the inclusion of a set $\mathcal{B}$ of *bad inputs*. For inputs in this set, data stability requirements do not necessarily hold. We call the complement $\mathcal{B}^c = \mathcal{Z}^d \setminus \mathcal{B}$ the set of *good inputs*. The concept of good and bad inputs as well as related proof techniques were originally proposed by [38]. Here, we incorporate them into the more general concentration of measure formalism of [35]. A full proof of the following theorem is deferred to Appendix A.

**Theorem 4 (Moment-generating function (MGF) bound for good inputs).** *Let $\mathcal{B} \subseteq \mathcal{Z}^d$ be a measurable set of bad inputs. Suppose for each $i \in [d+1]$, $V^{(i)}$ is a Wasserstein matrix for the*

Markov kernel $K^{(i)}$ defined in (7) on the set of good inputs, that is

$$\delta_k(K^{(i)}\widetilde{f}) \leq \sum_{j \in [d]} V_{kj}^{(i)} \delta_j(\widetilde{f}), \qquad \forall k \in [d] \tag{11}$$

for all Lipschitz (with respect to $\rho^d$) functions $\widetilde{f} \colon \mathcal{B}^c \to \mathbb{R}$. Define the Wasserstein dependency matrix

$$\Gamma \in \mathbb{R}^{d \times d}, \qquad \Gamma_{ij} = \|\rho\| V_{ij}^{(i+1)} \tag{12}$$

Then for all Lipschitz functions $f \colon \mathcal{Z}^d \to \mathbb{R}$, the following MGF bound holds

$$\mathbb{E}_{z \sim \mu|\mathcal{B}^c} \left[ \exp\left( \lambda(f(z) - \mathbb{E}_{\mu|\mathcal{B}^c} f) \right) \right] \leq \exp\left( \frac{\lambda^2}{8} \|\Gamma\delta(f)\|_2^2 \right). \tag{13}$$

An upper bound on the moment generating function will be used in the PAC-Bayesian argument concluding this section. The function $f$ in question will be the loss of a structured datum $z$. Regarding (13), our goal is to bound the norm $\|\Gamma\delta(f)\|_2^2$ through properties of the data distribution. We will use the fact that data is represented by measure transport to establish such a bound after the following preparatory lemma.

**Lemma 5 (Coupling from transport).** *Let $\nu^d$ be a reference measure on $\mathcal{Z}^d$ and $F, G \colon \mathcal{Z}^d \to \mathcal{Z}^d$ be measurable maps. Define the map $(F, G)$ by*

$$(F, G) \colon \mathcal{Z}^d \to \mathcal{Z}^d \times \mathcal{Z}^d, \qquad z \mapsto (F(z), G(z)) \tag{14}$$

*Then $(F, G)_\sharp \nu^d$ is a coupling of $F_\sharp \nu^d$ and $G_\sharp \nu^d$.*

*Proof.* Let $A \subseteq \mathcal{Z}^d$ be measurable, then

$$(F, G)_\sharp \nu^d(A, \mathcal{Z}^d) = \nu^d((F, G)^{-1}(A, \mathcal{Z}^d)) = \nu^d(F^{-1}(A)) = F_\sharp \nu^d(A) \tag{15}$$

which shows that $F_\sharp \nu^d$ is the first marginal of $(F, G)_\sharp \nu^d$. An analogous argument for the second marginal shows the assertion. □

By assuming $\mu$ to be represented via KR-rearrangement of a factorizing reference measure, Lemma 3 gives an explicit representation of KR-rearrangement for conditional distributions. From there, we invoke Lemma 5 to construct a coupling between conditional distributions and subsequently follow a line of reasoning put forward in [35] to explicitly construct Wasserstein matrices for the kernels (7) which yield a bound on (13) by Theorem 4. This leads to the following proposition. A full proof is deferred to Appendix A.

**Proposition 6 (Wasserstein dependency matrix from KR-rearrangement).** *Let $(\mathcal{Z}^d, \Sigma, \mu)$ be a probability space with $\mu = T_\sharp \nu^d$ for the KR-rearrangement $T \colon \mathcal{Z}^d \to \mathcal{Z}^d$ and a reference measure $\nu^d$ on $\mathcal{Z}^d$. Let each $\mathcal{Z}$ be equipped with a metric $\rho$ and have finite diameter $\|\rho\| < \infty$. Let $f \colon \mathcal{Z}^d \to \mathbb{R}$ be a Lipschitz function with respect to the product metric $\rho^d$. Let $\mathcal{B} \subseteq \mathcal{Z}^d$ denote a set of bad inputs and define the corresponding set $\mathcal{A} = T^{-1}(\mathcal{B}) \subseteq \mathcal{Z}^d$. Let $\widehat{T}$ be the unique KR-rearrangement that satisfies $\widehat{T}_\sharp \nu^d = \nu^d|\mathcal{A}^c$ and denote $\widetilde{T} = T \circ \widehat{T}$. Suppose there exist constants $L_{ij}$ such that for all $v, z \in \mathcal{B}^c$ with $v^{[d]\setminus\{i\}} = z^{[d]\setminus\{i\}}$ it holds*

$$\mathbb{E}_{\tau \sim \nu^{(i,d]}} \left[ \rho(\widetilde{T}^{(i,d]}(\widehat{v}^{[i]}, \tau)_j, \widetilde{T}^{(i,d]}(\widehat{z}^{[i]}, \tau)_j) \right] \leq L_{ij} \rho(v_i, z_i) \tag{16}$$

*where $\widehat{v}^{[i]}$ and $\widehat{z}^{[i]}$ are uniquely defined through $\widetilde{T}^{[i]}(\widehat{v}^{[i]}) = v^{[i]}$ and $\widetilde{T}^{[i]}(\widehat{z}^{[i]}) = z^{[i]}$. Then $\Gamma = \frac{\|\rho\|}{d} D$ is a Wasserstein dependency matrix for $\mu|\mathcal{B}^c$ with*

$$D_{ij} = \begin{cases} 0 & \text{if } i > j, \\ 1 & \text{if } i = j, \\ L_{ij} & \text{if } i < j. \end{cases} \tag{17}$$

We remark that $\Gamma$ indeed distills the dependency structure of $\mu$. To illustrate this, let $\mu^{\{i\}}$ be independent from $\mu^{\{j\}}$ for some $i \in [d]$, $j \in (i, d]$. Then conditioning on a different value of $\mu^{\{i\}}$ does not change the distribution $\mu^{\{j\}}$. Thus,

$$\widetilde{T}^{(i,d]}(\overline{x}^{[i]}, \tau)_j = \widetilde{T}^{(i,d]}(\overline{z}^{[i]}, \tau)_j, \qquad \forall \tau \in \mathcal{Z}^{(i,d]}, \quad \overline{x}^{[d]\setminus\{i\}} = \overline{z}^{[d]\setminus\{i\}}, \quad j \in (i, d] \tag{18}$$

and the choice $L_{ij} = 0$ satisfies (16). It directly follows that $\Gamma_{ij} = 0$.

The following theorem states the main result of the present work.

**Theorem 7** (PAC-Bayesian risk certificate for structured prediction). *Fix $\delta \in (0, \exp(-e^{-1}))$, let $\mu$ be a data distribution on $\mathcal{Z}^d$ with $T_\sharp \nu^d = \mu$ the Knothe-Rosenblatt rearrangement for a reference measure $\nu^d$ on $\mathcal{Z}^d$ and fix a measurable set $\mathcal{B} \subseteq \mathcal{Z}^d$ of bad inputs with $\mu(\mathcal{B}) \leq \xi$. Fix a PAC-Bayes prior $\pi$ on a hypothesis class $\mathcal{H}$ of functions $\phi \colon \mathcal{X}^d \to \mathcal{Y}^d$ and a loss function $\ell$ which assumes values in $[0,1]$. Define the oscillation vector $\widetilde{\delta}$ by*

$$\widetilde{\delta}_i = \sup_{h \in \mathcal{H}} \delta_i\big( L(h, \cdot)\big|_{\mathcal{B}^c}\big), \qquad i \in [d] \tag{19}$$

*where $L(h, \cdot)\big|_{\mathcal{B}^c}$ denotes the restriction of $L(h, \cdot)$ to $\mathcal{Z}^d \setminus \mathcal{B}$. Suppose all oscillations $\widetilde{\delta}_i$ are finite, suppose the condition (16) is satisfied and denote by $D$ the matrix with entries (17). Then, with probability at least $1 - \delta$ over realizations of a training set $\mathcal{D}_m = (Z^{(k)})_{k=1}^m$ drawn from $(\mu|\mathcal{B}^c)^m$ it holds for all PAC-Bayes posteriors $\zeta$ on $\mathcal{H}$ that*

$$\mathcal{R}(\zeta) \leq \mathcal{R}_m(\zeta, \mathcal{D}_m) + 2\frac{\|\rho\|}{d}\|D\widetilde{\delta}\|_2 \sqrt{\frac{\log \frac{1}{\delta} + \mathrm{KL}[\zeta : \pi]}{2m}} + \xi. \tag{20}$$

Note that the generalization gap on the right hand side of (20) decays with $d$. This accounts for generalization from a *single* example. In fact, if only $m = 1$ structured example is available, but $d \gg 1$, Theorem 7 still certifies risk. This effect can however be negated by the norm $\|D\widetilde{\delta}\|$. If structured data contain strong global dependence, then $\|D\widetilde{\delta}\|$ will not be bounded independently of $d$ and thus, in the worst case of $\|D\widetilde{\delta}\| \in \mathcal{O}(d)$ the assertion is no stronger than PAC-Bayesian bounds for unstructured data. The same point was observed in [38].

The measure of the bad set under the data distribution $\mu$ is assumed to be bounded by $\xi$. This is to account for a small number of data which contain strong dependence. In order to prevent these bad data from dominating $D$, thereby negating the decay of the bound in $d$ as described above, it is preferable to exclude them from the sample, reduce the sample size $m$ and pay the penalty $\xi$ in (20).

To prove Theorem 7, we broadly follow the argument put forward in [38]. This augments typical PAC-Bayesian constructions in the literature by the inclusion of a set of bad inputs. We first reconcile the data being conditioned on $\mathcal{B}^c$ with risk certification for unconditioned data, leading to the addition of $\xi$ on the right hand side of (20). The model complexity term $\mathrm{KL}[\zeta : \pi]$ is due to Donsker and Varadhand's variational formula. Subsequently, the moment generating function bound of Theorem 4 is instantiated through the Wasserstein dependency matrix constructed in Proposition 6. Markov's inequality then gives a pointwise risk bound for fixed value of a free parameter. In order to optimize this parameter, the bound is made uniform on a discrete set of values through a union bound. A full proof is presented in Appendix A.

In Theorem 7, we combine the PAC-Bayesian construction of [38] with the more general concentration of measure theory of [35]. Crucially, concentration of measure results used in [38] are predicated on the assumption of data generated by a Markov random field [57, 34]. Our work is more flexible in two major ways.

(1) Our assumption on the data-generating distribution is likely more representative of real-world data as measure transport models have repeatedly been shown to yield convincing data generators.

(2) Markov random fields are difficult to handle computationally because inference in general Markov random fields is NP-hard [57] such that one is forced to learn based on approximate inference procedures [31, 22, 55].

Our work also allows for more general metrics $\rho$ as opposed to the singular choice of Hamming norm required in [38]. Additionally, the key results of [38] are constructed to ensure all data drawn from the unconditioned distribution $\mu$ are in the good set. This reduces the probability of correctness $1 - \delta$ by $m\xi$. Instead, we assume data drawn from $\mu|\mathcal{B}^c$, effectively reducing the number of available samples by a factor of $1 - \xi$, but keeping the probability of correctness high. This allows the set of bad inputs to be used more effectively as a computational tool in Section 5.

Comparing the dependency of (20) on $d$ with the respective result in [38], it first appears as though our bound decays with a faster rate ($d$ instead of $\sqrt{d}$). However, this will not typically be the case in practice because $\|D\widetilde{\delta}\|_2$ grows with rate $\sqrt{d}$ in most situations. To see this, consider the case of local

dependency in the sense that

$$L_{ij} = \begin{cases} 1, & \text{if } j \in \mathcal{N}_i \, , \\ 0, & \text{else} \end{cases} \tag{21}$$

for local neighborhoods $\mathcal{N}_i \subseteq [d]$ which contain a fixed number of $c$ elements and let $\widetilde{\delta}_i = \alpha$ for all $i \in [d]$ and some constant value $\alpha > 0$. Then

$$\|D\widetilde{\delta}\|_2 = \sqrt{\sum_{i \in [d]} \left(\alpha|\mathcal{N}_i|\right)^2} = c\alpha\sqrt{d} \, . \tag{22}$$

Clearly, if dependence is localized and the oscillations $\widetilde{\delta}$ do not decay in $d$, then $\|D\widetilde{\delta}\|_2$ contains a factor that grows with rate $\sqrt{d}$, leading to the same asymptotic rate of (20) already observed in [38].

Note that [38] additionally allows for a set of bad hypotheses $\mathcal{B}_{\mathcal{F}} \subseteq \mathcal{H}$ which do not conform to stability assumptions. In our construction, this means restricting the bound (19) to oscillations on the set of good hypotheses. We omit this extension for clarity of exposition, but do not expect it to necessitate major changes to the presented proofs. The same applies to the derandomization strategy proposed by [38] which is based on hypothesis stability.

Further, [38] considers a large number of applicable orderings for random variables by introducing a filtration of their index set. This notion is not easily compatible with our assumption of KR-rearrangement, because triangularity of transport depends on the order of variables.

## 5   Bounding the Bad Set

With regard to numerical risk certificates, a key technical aspect of Section 4 concerns the quantities $L_{ij}$ in (16). Here, we propose a way to use the set of bad inputs as a computational tool to this end. Suppose we assign arbitrary fixed values to $L_{ij}$ and subsequently *define* $\mathcal{B} \subseteq \mathcal{Z}^d$ as the set of inputs on which the condition (16) fails. Then we have fulfilled the prerequisites of Proposition 6 by construction and are left with bounding $\mu(\mathcal{B})$. Note that

$$\mu(\mathcal{B}) = \mathbb{P}_{Z \sim \mu}(Z \in \mathcal{B}) = \mathbb{E}_{Z \sim \mu}[\mathbf{1}\{Z \in \mathcal{B}\}] \tag{23}$$

and the indicator function $\mathbf{1}$ assumes values in the bounded set $\{0, 1\}$. Therefore, Hoeffding's inequality gives the following.

**Proposition 8** (**Upper bound on the bad set**). *Let $\mu$ be a data distribution and let $\widetilde{\mathcal{D}}_n \sim \mu^n$ be a sample of size $n$. Fix an error probability $\epsilon \in (0, 1)$. Then*

$$\mu(\mathcal{B}) \leq \frac{1}{n} \sum_{Z \in \widetilde{\mathcal{D}}_n} \mathbf{1}\{Z \in \mathcal{B}\} + \sqrt{\frac{1}{2n} \log \frac{2}{\epsilon}} \tag{24}$$

*with probability at least $1 - \epsilon$ over the sample.*

Checking the condition $Z \in \mathcal{B}$ requires evaluating (16) which comes down to finding a Lipschitz constant for a one-dimensional function. If this is computationally feasible for the given data model, then the concentration argument (24) can be used to bound $\mu(\mathcal{B})$ with high probability. Because (24) decays only in the number of structured examples $\mathcal{O}(\sqrt{n})$, it can not be used to show generalization from a single structured example. However, PAC-Bayesian risk certificates are typically dominated by the KL complexity term in (20) which decays with the size of structured examples as well. Thus, Proposition 8 should still be useful in practice.

Note that Proposition 8 makes a pointwise statement about a fixed value of $L_{ij}$ which has limited utility for learning $L_{ij}$ from data. To remedy this problem, we can first define a discrete set $\mathcal{L} = (L^{(k)})_{k \in [l]}$ of candidate matrices and select error probabilities $\epsilon_k$ for each of the events

$$\mu(\mathcal{B}(L^{(k)})) \geq \frac{1}{n} \sum_{Z \in \widetilde{\mathcal{D}}_n} \mathbf{1}\{Z \in \mathcal{B}(L^{(k)})\} + \sqrt{\frac{1}{2n} \log \frac{2}{\epsilon_k}} \tag{25}$$

such that $\sum_{k \in [l]} \epsilon_k = \epsilon$. Then, by a union bound with probability at least $1 - \epsilon$ over the sample none of the events (25) occurs. We have thus constructed a uniform bound over the set of candidate matrices which allows us to select the one which minimizes the generalization gap in (20).

In order to make this strategy most effective, domain knowledge on the application at hand should be applied when constructing candidate matrices and assigning error probabilities. For instance, the limited empirical findings of [14] on ImageNet [17] indicate that the majority of natural images contain mostly local signal. In image segmentation, this is conducive to concentration, because it can lead to many small values in an optimal Wasserstein dependency matrix. In particular, if dependency decays with distance in the image domain, one should select configurations $L^{(k)}$ in which $L_{ij}^{(k)}$ is small if $i$ is distant from $j$ in the image domain and allowed to assume larger values if $i$ is close to $j$ in the image domain.

We give an *intuitive interpretation* of the relationship between Proposition 8 and Theorem 7 as follows. Suppose the majority of samples from a structured data distribution contain mostly local signal. The locality of signal in samples indicates weak global dependence of random variables which in turn manifests in small entries of a Wasserstein dependency matrix. However, a small number of bad data may contain only weak local signal. For instance, an image in which every pixel has the same value does not give more information to a learner if it is doubled in size. Even worse, a small (but not null) set of bad data will dominate the Wasserstein dependency matrix and prevent generalization that scales with $d$. Proposition 8 thus estimates an upper bound on the likelihood of bad data under $\mu$ which are then excluded from the concentration argument underlying the bound (20). This relationship is further illustrated in a numerical toy example in Appendix C.

## 6 Limitations

We assume that structured data are drawn from a distribution $\mu$ which arises from a reference measure through KR-rearrangement. This is motivated by the fact that a large class of data distributions can be represented in this way and that some assumption on the data-generating distribution is required in order to make statements on conditional distributions required to quantify dependence. However, it is an open question how closely a measure transport distribution learned from data approximates the unknown distribution from which the data are drawn. A first step towards this goal is made in the recent work [4], but a non-asymptotic theory of generalization for generative models is left for future work. Accordingly, we do not present any empirical results on real-world data.

We describe how dependency in the data distribution which is relevant for generalization of discriminative models can be written as properties of the KR-rearrangement through a Wasserstein dependency matrix. Section 5 further outlines how the construction of bad data can be used as a computational tool to this end. We do not cover how to compute the involved Lipschitz constants $L_{ij}$ in (16). In practice, this will require nontrivial numerical machinery because a deterministic bound on the expected value under the reference distribution appears needed. One possible approach is to employ quasi-Monte-Carlo methods [19, 18] which come with deterministic error bounds and have recently been applied to PAC-Bayesian certification of ordinary (non-structured) classification risk [7]. This entails a fine-grained stability analysis of the involved integrands and is likely to be computationally expensive because deterministic error bounds tend to be pessimistic compared to their stochastic counterparts.

## 7 Conclusion

We have presented a PAC-Bayesian risk certificate for structured prediction. Compared to earlier work, we make the assumption of data generated by KR-rearrangement of a reference measure. This approach is able to represent all atom-free data distributions and yields an explicit quantification of dependence via Wasserstein dependency matrices. It also indicates a way of leveraging powerful generative models to compute risk certificates for downstream discriminative tasks.

### Acknowledgments and Disclosure of Funding

This work is funded by the Deutsche Forschungsgemeinschaft (DFG), grant SCHN 457/17-1, within the priority programme SPP 2298: "Theoretical Foundations of Deep Learning". This work is funded by the Deutsche Forschungsgemeinschaft (DFG) under Germany's Excellence Strategy EXC-2181/1 - 390900948 (the Heidelberg STRUCTURES Excellence Cluster).

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

# A  Full Proofs of Presented Results

In this appendix, we present full proofs of all results which are not already complete in the main text.

**Proof of Theorem 4**   For any $i \in [d]$ and $z \in \mathcal{Z}^d$ define

$$M^{(i)} = \mathbb{E}_{Z \sim \mu}[f(Z)|\mathcal{B}^c, Z^{[i]} = z^{[i]}] - \mathbb{E}_{Z \sim \mu}[f(Z)|\mathcal{B}^c, Z^{[i-1]} = z^{[i-1]}] \tag{26}$$

with the edge case

$$M^{(1)} = \mathbb{E}_{Z \sim \mu}[f(Z)|\mathcal{B}^c, Z_1 = z_1] - \mathbb{E}_{Z \sim \mu}[f(Z)|\mathcal{B}^c] . \tag{27}$$

Due to $\mathbb{E}_{Z \sim \mu}[f(Z)|\mathcal{B}^c, Z = z] = f(z)$ for $z \in \mathcal{B}^c$ we have

$$f - \mathbb{E}_{Z \sim \mu}[f(Z)|\mathcal{B}^c] = \sum_{i=1}^{d} M^{(i)} . \tag{28}$$

Since the conditions $Z^{[i]} = z^{[i]}$ generate a nested sequence of $\sigma$-algebras, the quantities $K^{(i+1)}f(z) = \mathbb{E}_\mu[f(Z)|\mathcal{B}^c, Z^{[i]} = z^{[i]}]$ are a Doob martingale and (26) is a martingale difference sequence. In order to bound the moment generating function of $f$, we will bound every $M^{(i)}$ from above and below and apply the Azuma-Hoeffding theorem 10. We have

$$M^{(i)} = \mathbb{E}_\mu[f(Z)|\mathcal{B}^c, Z^{[i]} = z^{[i]}] - \mathbb{E}_\mu[f(Z)|\mathcal{B}^c, Z^{[i-1]} = z^{[i-1]}] \tag{29a}$$

$$= \mathbb{E}_\mu[f(Z)|\mathcal{B}^c, Z^{[i]} = z^{[i]}] - \mathbb{E}_\mu[\mathbb{E}_\mu[f(Z)|\mathcal{B}^c, Z^{[i-1]} = z^{[i-1]}, Z_i]|\mathcal{B}^c, Z^{[i-1]} = z^{[i-1]}] \tag{29b}$$

$$= \int f(z^{[i]}w^{(i,d)})\mu(dw^{(i,d)}|z^{[i]}, \mathcal{B}^c)$$
$$- \int \left( \int f(z^{[i]}u^{(i,d)})\mu(du^{(i,d)}|z^{[i-1]}, w_i, \mathcal{B}^c) \right)\mu(dw^{[i,d]}|z^{[i-1]}, \mathcal{B}^c) \tag{29c}$$

by the tower property of conditional expectations. Because $\mu(dw^{[i,d]}|z^{[i-1]}, \mathcal{B}^c)$ is a probability measure, it holds

$$\int f(z^{[i]}w^{(i,d)})\mu(dw^{(i,d)}|z^{[i]}, \mathcal{B}^c) = \int \left( \int f(z^{[i-1]}z_i u^{(i,d)})\mu(du^{(i,d)}|z^{[i]}, \mathcal{B}^c) \right)\mu(dw^{[i,d]}|z^{[i-1]}, \mathcal{B}^c) \tag{30}$$

and we find

$$M^{(i)} = \int \mu(dw^{[i,d]}|z^{[i-1]}, \mathcal{B}^c)\left( \int f(z^{[i-1]}z_i u^{(i,d)})\mu(du^{(i,d)}|z^{[i]}, \mathcal{B}^c) \right.$$
$$\left. - \int f(z^{[i]}u^{(i,d)})\mu(du^{(i,d)}|z^{[i-1]}, w_i, \mathcal{B}^c) \right) \tag{31}$$

Now bound $A^{(i)} \leq M^{(i)} \leq B^{(i)}$ almost surely with

$$A^{(i)} = \int \mu(dw^{[i,d]}|z^{[i-1]}, \mathcal{B}^c) \inf_{z_i \in \mathcal{B}_i^c(z^{[i-1]})} \left( \int f(z^{[i-1]}z_i u^{(i,d)})\mu(du^{(i,d)}|z^{[i]}, \mathcal{B}^c) \right.$$
$$\left. - \int f(z^{[i]}u^{(i,d)})\mu(du^{(i,d)}|z^{[i-1]}, w_i, \mathcal{B}^c) \right) \tag{32a}$$

$$B^{(i)} = \int \mu(dw^{[i,d]}|z^{[i-1]}, \mathcal{B}^c) \sup_{z_i \in \mathcal{B}_i^c(z^{[i-1]})} \left( \int f(z^{[i-1]}z_i u^{(i,d)})\mu(du^{(i,d)}|z^{[i]}, \mathcal{B}^c) \right.$$
$$\left. - \int f(z^{[i]}u^{(i,d)})\mu(du^{(i,d)}|z^{[i-1]}, w_i, \mathcal{B}^c) \right) \tag{32b}$$

where $\mathcal{B}_i^c(z^{[i-1]})$ contains all $z_i \in \mathcal{Z}$ such that there exist $z^{(i,d]} \in \mathcal{Z}^{d-i}$ with $(z^{[i-1]}, z_i, z^{(i,d]}) \in \mathcal{B}^c$. Because every realization of a random variable conditioned on $\mathcal{B}^c$ is in the set of good inputs, the difference $\|B^{(i)} - A^{(i)}\|_\infty$ can be written as

$$\sup_{v, z \in \mathcal{B}^c, v^{[d] \setminus \{i\}} = z^{[d] \setminus \{i\}}} \int f(v^{[i]}u^{(i,d)})\mu(du^{(i,d)}|v^{[i]}, \mathcal{B}^c) - \int f(z^{[i]}u^{(i,d)})\mu(du^{(i,d)}|z^{[i]}, \mathcal{B}^c) \tag{33}$$

By seeing this expression in terms of oscillation of the kernel action $K^{(i+1)}f$, we find

$$\|B^{(i)} - A^{(i)}\|_\infty \le \|\rho\|\delta_i(K^{(i+1)}\widetilde{f}) \le \|\rho\|(V^{(i+1)}\delta(\widetilde{f}))_i = (\Gamma\delta(\widetilde{f}))_i \tag{34}$$

where $\widetilde{f}\colon \mathcal{B}^c \to \mathbb{R}$ is the restriction of $f$ to $\mathcal{B}^c$. The assertion then follows from the Azuma-Hoeffding theorem [35, Theorem 4.1] which we recite as Theorem 10 to make this paper self-contained.

**Proof of Proposition 6**   For arbitrary $z, z' \in \mathcal{Z}^d$ it holds

$$|f(z) - f(z')| \le \delta_j(f)\rho(z_j, z'_j), \qquad \forall\, i \in [d] \tag{35}$$

and thus, by summing over all indices we get

$$|f(z) - f(z')| \le \frac{1}{d}\sum_{j\in[d]} \delta_j(f)\rho(z_j, z'_j) \tag{36}$$

Let $v, z \in \mathcal{Z}^d$ with $v^{[d]\setminus\{i\}} = z^{[d]\setminus\{i\}}$ be given for some $i \in [d]$. Recall the action (8) of Markov kernels $K^{(i+1)}$ is an expected value with respect to conditional distributions $\mu^{(i,d)}(dw^{(i,d)}|v^{[i]})$.

Because $\nu^d$ has no atoms, $\nu^d|\mathcal{A}^c$ also has no atoms. Therefore, there is a unique KR-rearrangement $\widehat{T}$ with $\widehat{T}_\sharp \nu^d = \nu^d|\mathcal{A}^c$. Then $\widetilde{T} = T \circ \widehat{T}$ is a KR-rearrangement with

$$\widetilde{T}_\sharp \nu^d = \mu|\mathcal{B}^c \tag{37}$$

by Lemma 9 and we have $\widetilde{T}(\widehat{v}) = v$. Lemma 3 implies

$$\mu^{(i,d)}(dw^{(i,d)}|\mathcal{B}^c, v^{[i]}) = \widetilde{T}(\widehat{v}^{[i]}, \cdot)_\sharp \nu^{d-i} \tag{38}$$

An analogous expression holds for the distribution conditioned on $z$. We have therefore found two transport functions pushing the reference measure to the respective conditional distributions. By Lemma 5, a coupling of the conditional distributions is then given by

$$P_{v,z}^{[i]} = (\widetilde{T}^{(i,d)}(\widehat{v}^{[i]}, \cdot), \widetilde{T}^{(i,d)}(\widehat{z}^{[i]}, \cdot))_\sharp \nu^{d-i} . \tag{39}$$

Using a change of measure we find

$$K^{(i+1)}f(v) - K^{(i+1)}f(z)$$

$$= \int P_{v,z}^{[i]}(du^{(i,d)}, dv^{(i,d)})\big(f(v^{[i]}u^{(i,d)}) - f(z^{[i]}v^{(i,d)})\big) \tag{40}$$

$$= \int \big(f(v^{[i]}\widetilde{T}^{(i,d)}(\widehat{v}^{[i]}, \tau)) - f(z^{[i]}\widetilde{T}^{(i,d)}(\widehat{z}^{[i]}, \tau))\big)\nu^{d-i}(\tau) \tag{41}$$

$$\le \frac{\delta_i(f)}{d}\rho(v_i, z_i) + \sum_{j\in(i,d]} \frac{\delta_j(f)}{d}\int \rho\big(\widetilde{T}^{(i,d)}(\widehat{v}^{[i]}, \tau)_j, \widetilde{T}^{(i,d)}(\widehat{z}^{[i]}, \tau)_j\big)\nu^{d-i}(\tau) \tag{42}$$

$$\le \frac{\delta_i(f)}{d}\rho(v_i, z_i) + \sum_{j\in(i,d]} \frac{\delta_j(f)}{d}L_{ij}\rho(v_i, z_i) \tag{43}$$

which shows

$$\delta_i(K^{(i+1)}f) \le \frac{1}{d}\Big(\delta_i(f) + \sum_{j\in(i,d]} L_{ij}\delta_j(f)\Big) \tag{44}$$

for good inputs. We have thus found a Wasserstein matrix $V^{(i+1)}$ for $K^{(i+1)}$ with entries

$$V_{ij}^{(i+1)} = \begin{cases} 0 & \text{if } i > j \\ d^{-1} & \text{if } i = j \\ d^{-1}L_{ij} & \text{if } i < j \end{cases} \tag{45}$$

in row $i$ which shows the assertion.

**Proof of Theorem 7** For any hypothesis $h \in \mathcal{H}$, we have

$$\mathcal{R}(h) - \mathcal{R}_m(h, \mathcal{D}_m) = \mathbb{E}_{Z \sim \mu}[L(h, Z) - \mathcal{R}_m(h, \mathcal{D}_m)] \tag{46a}$$

$$= \mathbb{E}_{Z \sim \mu}\left[\left(L(h, Z) - \mathcal{R}_m(h, \mathcal{D}_m)\right)\mathbf{1}\{Z \notin \mathcal{B}\}\right]$$
$$+ \mathbb{E}_{Z \sim \mu}\left[\left(L(h, Z) - \mathcal{R}_m(h, \mathcal{D}_m)\right)\mathbf{1}\{Z \in \mathcal{B}\}\right] \tag{46b}$$

$$\leq \mathbb{E}_{Z \sim \mu}\left[\left(L(h, Z) - \mathcal{R}_m(h, \mathcal{D}_m)\right)\mathbf{1}\{Z \notin \mathcal{B}\}\right] + \xi \tag{46c}$$

$$\leq \mathbb{E}_{Z \sim \mu|\mathcal{B}^c}[L(h, Z)] - \mathcal{R}_m(h, \mathcal{D}_m) + \xi \tag{46d}$$

where in (46c) we have used that pointwise loss is in $[0, 1]$. Note that the underlying distribution of the risk $\mathcal{R}(h)$ is $\mu$, while $\mathcal{D}_m$ are drawn from $\mu|\mathcal{B}^c$. The above inequality reconciles this such that a concentration argument for the conditional distribution becomes applicable. For any PAC-Bayes posterior distribution $\zeta$ and any $\beta > 0$, this implies

$$\mathcal{R}(\zeta) - \mathcal{R}_m(\zeta, \mathcal{D}_m) = \mathbb{E}_{h \sim \zeta}\mathbb{E}_{Z \sim \mu}[L(h, Z) - \mathcal{R}_m(h, \mathcal{D}_m)] \tag{47a}$$

$$\leq \mathbb{E}_{h \sim \zeta}\left[\mathbb{E}_{Z \sim \mu|\mathcal{B}^c}[L(h, Z)] - \mathcal{R}_m(h, \mathcal{D}_m)\right] + \xi \tag{47b}$$

$$= \frac{1}{\beta}\mathbb{E}_{h \sim \zeta}\left[\beta(\mathbb{E}_{Z \sim \mu|\mathcal{B}^c}[L(h, Z)] - \mathcal{R}_m(h, \mathcal{D}_m))\right] + \xi \tag{47c}$$

$$\leq \frac{1}{\beta}\log\mathbb{E}_{h \sim \pi}\left[\exp\left(\beta(\mathbb{E}_{Z \sim \mu|\mathcal{B}^c}[L(h, Z)] - \mathcal{R}_m(h, \mathcal{D}_m))\right)\right]$$
$$+ \frac{1}{\beta}\mathrm{KL}[\zeta : \pi] + \xi \tag{47d}$$

by Donsker and Varadhan's variational formula [2, Lemma 2.2]. Focusing on the first term, we find

$$\exp\left(\beta(\mathbb{E}_{Z \sim \mu|\mathcal{B}^c}[L(h, Z)] - \mathcal{R}_m(h, \mathcal{D}_m))\right) = \exp\left(\frac{\beta}{m}\sum_{k \in [m]}\left(\mathbb{E}_{Z \sim \mu|\mathcal{B}^c}[L(h, Z)] - L(h, Z^{(k)})\right)\right) \tag{48a}$$

$$= \prod_{k \in [m]}\exp\left(\frac{\beta}{m}\left(\mathbb{E}_{Z \sim \mu|\mathcal{B}^c}[L(h, Z)] - L(h, Z^{(k)})\right)\right) \tag{48b}$$

Each structured datum $Z^{(k)}$ is drawn independently from $\mu|\mathcal{B}^c$. By Proposition 6 there exists a Wasserstein dependency matrix $\Gamma = \frac{\|\rho\|}{d}D$ for $\mu|\mathcal{B}^c$ where $D$ has entries (17). Then

$$\mathbb{E}_{\mathcal{D}_m \sim (\mu|\mathcal{B}^c)^m}\prod_{k \in [m]}\exp\left(\frac{\beta}{m}\left(\mathbb{E}_{Z \sim \mu|\mathcal{B}^c}[L(h, Z)] - L(h, Z^{(k)})\right)\right)$$

$$= \prod_{k \in [m]}\mathbb{E}_{Z^{(k)} \sim (\mu|\mathcal{B}^c)}\exp\left(\frac{\beta}{m}\left(\mathbb{E}_{Z \sim \mu|\mathcal{B}^c}[L(h, Z)] - L(h, Z^{(k)})\right)\right) \tag{49a}$$

$$= \prod_{k \in [m]}\mathbb{E}_{Z^{(k)} \sim \mu|\mathcal{B}^c}\left[\exp\left(\frac{\beta}{m}\left(\mathbb{E}_{Z \sim \mu|\mathcal{B}^c}[L(h, Z)] - L(h, Z^{(k)})\right)\right)\right] \tag{49b}$$

$$\leq \prod_{k \in [m]}\exp\left(\frac{\beta^2}{8m^2}\|\Gamma\delta(\widetilde{L}(h, \cdot))\|_2^2\right) \text{ by Theorem 4} \tag{49c}$$

$$= \exp\left(\frac{\beta^2}{8m}\|\Gamma\delta(\widetilde{L}(h, \cdot))\|_2^2\right) \tag{49d}$$

$$\leq \exp\left(\frac{\beta^2}{8m}\|\Gamma\widetilde{\delta}\|_2^2\right) \tag{49e}$$

Denote the shorthand

$$U = \mathbb{E}_{\mathcal{D}_m \sim (\mu|\mathcal{B}^c)^m}\left[\exp\left(\beta(\mathbb{E}_{Z \sim \mu|\mathcal{B}^c}[L(h, Z)] - \mathcal{R}_m(h, \mathcal{D}_m))\right)\right] \tag{50}$$

By Markov's inequality it holds

$$\mathbb{P}_{\mathcal{D}_m \sim (\mu|\mathcal{B}^c)^m} \left[ \exp\left( \beta(\mathbb{E}_{Z \sim \mu|\mathcal{B}^c}[L(h, Z)] - \mathcal{R}_m(h, \mathcal{D}_m)) \right) \geq \frac{1}{\delta} U \right] \leq \delta \tag{51}$$

and combining this with (49) we have

$$\exp\left( \beta(\mathbb{E}_{Z \sim \mu|\mathcal{B}^c}[L(h, Z)] - \mathcal{R}_m(h, \mathcal{D}_m)) \right) \leq \frac{1}{\delta} \exp\left( \frac{\beta^2}{8m} \|\Gamma \widetilde{\delta}\|_2^2 \right) \tag{52}$$

with probability at least $1 - \delta$ over the sample. Using (47) we thus have

$$\mathcal{R}(\zeta) - \mathcal{R}_m(\zeta, \mathcal{D}_m) \leq \frac{1}{\beta} \left( \log \mathbb{E}_{h \sim \pi} \left[ \frac{1}{\delta} \exp\left( \frac{\beta^2}{8m} \|\Gamma \widetilde{\delta}\|_2^2 \right) \right] + \mathrm{KL}[\zeta : \pi] \right) + \xi \tag{53a}$$

$$= \frac{\beta}{8m} \|\Gamma \widetilde{\delta}\|_2^2 + \frac{1}{\beta} \left( \log \frac{1}{\delta} + \mathrm{KL}[\zeta : \pi] \right) + \xi \tag{53b}$$

Ideally, we would minimize the right hand side with respect to $\beta$. However, this would mean to have $\beta$ depend on $\zeta$ and we thus would not have a uniform bound for all posterior distributions.

Instead, [38] approaches the problem by defining a sequence of constant $(\delta_j, \beta_j)_{j \in \mathbb{N}_0}$ and bounding the probability that the bound does not hold for any sequence element. Since in the opposite (high-probability) case, the bound holds for all sequence elements, an optimal one can subsequently be chosen dependent on the posterior.

For all $j \in \mathbb{N}_0$, define

$$\delta_j = \delta 2^{-(j+1)}, \qquad \beta_j = 2^j \sqrt{\frac{8m \log \frac{1}{\delta}}{\|\Gamma \widetilde{\delta}\|_2^2}} \tag{54}$$

which are independent of $\zeta$. Now consider the event $E_j$ that

$$\exp\left( \beta_j(\mathbb{E}_{Z \sim \mu|\mathcal{B}^c}[\ell(h, Z)] - \mathcal{R}_m(h, \mathcal{D}_m)) \right) \geq \frac{1}{\delta_j} \exp\left( \frac{\beta_j^2}{8m} \|\Gamma \widetilde{\delta}\|_2^2 \right) \tag{55}$$

By the above argument leading up to (52), the probability for $E_j$ under a random sample of the conditioned data distribution $\mu|\mathcal{B}^c$ is at most $\delta_j$. Therefore, the probability that any $E_j$ occurs is bounded by

$$\mathbb{P}\left( \bigcup_{j \in \mathbb{N}_0} E_j \right) \leq \sum_{j \in \mathbb{N}_0} \mathbb{P}(E_j) \leq \sum_{j \in \mathbb{N}_0} \delta_j = \delta \tag{56}$$

Thus, for all posteriors $\zeta$ with probability at least $1 - \delta$ none of the events (55) occurs. We may therefore select an index $j$ dependent on $\zeta$ to obtain a sharper risk certificate which still holds with probability at least $1 - \delta$ over the sample conditioned on the good set. For a fixed posterior $\zeta$, the optimizer of (53b) would be

$$\beta^* = \frac{1}{\|\Gamma \widetilde{\delta}\|_2} \sqrt{8m(\log \frac{1}{\delta} + \mathrm{KL}[\zeta : \pi])} \tag{57}$$

Equating this to (55) and rounding down to the nearest integer gives

$$j^* = \left\lfloor \frac{1}{2} \log_2 \left( 1 + \frac{\mathrm{KL}[\zeta : \pi]}{\log \frac{1}{\delta}} \right) \right\rfloor \tag{58}$$

Denote this number before rounding by $r$, i.e. $j^* = \lfloor r \rfloor$. For any real number $r$ it holds $r - 1 \leq \lfloor r \rfloor \leq r$. Therefore

$$\frac{1}{2} \sqrt{1 + \frac{\mathrm{KL}[\zeta : \pi]}{\log \frac{1}{\delta}}} = 2^{r-1} \leq 2^{j^*} \leq 2^r = \sqrt{1 + \frac{\mathrm{KL}[\zeta : \pi]}{\log \frac{1}{\delta}}} \tag{59}$$

which gives the following bounds on $u_{j^*}$

$$\frac{1}{2} \sqrt{\frac{8m(\log \frac{1}{\delta} + \mathrm{KL}[\zeta : \pi])}{\|\Gamma \widetilde{\delta}\|_2^2}} \leq u_{j^*} \leq \sqrt{\frac{8m(\log \frac{1}{\delta} + \mathrm{KL}[\zeta : \pi])}{\|\Gamma \widetilde{\delta}\|_2^2}} \tag{60}$$

Likewise, we bound

$$\mathrm{KL}[\zeta:\pi] + \log\frac{1}{\delta_{j^*}} = \mathrm{KL}[\zeta:\pi] + \log\frac{2}{\delta} + j^*\log 2 \tag{61a}$$

$$\leq \mathrm{KL}[\zeta:\pi] + \log\frac{2}{\delta} + \frac{\log 2}{2}\log_2\left(1 + \frac{\mathrm{KL}[\zeta:\pi]}{\log\frac{1}{\delta}}\right) - \log 2 \tag{61b}$$

$$= \mathrm{KL}[\zeta:\pi] + \log\frac{1}{\delta} + \frac{1}{2}\log\left(1 + \frac{\mathrm{KL}[\zeta:\pi]}{\log\frac{1}{\delta}}\right) \tag{61c}$$

$$= \mathrm{KL}[\zeta:\pi] + \log\frac{1}{\delta} + \frac{1}{2}\log\left(\log\frac{1}{\delta} + \mathrm{KL}[\zeta:\pi]\right) - \frac{1}{2}\log\log\frac{1}{\delta} \tag{61d}$$

The assumption $\delta \leq \exp(-e^{-1})$ yields $-\log\log\frac{1}{\delta} \leq 1$ and because $x + 1 \leq \exp(x)$ for all $x \in \mathbb{R}$, we find

$$\mathrm{KL}[\zeta:\pi] + \log\frac{1}{\delta_{j^*}} \leq \mathrm{KL}[\zeta:\pi] + \log\frac{1}{\delta} + \frac{1}{2}\left(\log\left(\log\frac{1}{\delta} + \mathrm{KL}[\zeta:\pi]\right) + 1\right) \tag{62a}$$

$$\leq \mathrm{KL}[\zeta:\pi] + \log\frac{1}{\delta} + \frac{1}{2}\left(\log\frac{1}{\delta} + \mathrm{KL}[\zeta:\pi]\right) \tag{62b}$$

$$= \frac{3}{2}\left(\log\frac{1}{\delta} + \mathrm{KL}[\zeta:\pi]\right) \tag{62c}$$

We can now use the bounds (62c) and (60) in (53b) to bound the expected generalization error

$$\mathcal{R}(\zeta) - \mathcal{R}_m(\zeta, \mathcal{D}_m) \leq \frac{u_{j^*}}{8m}\|\Gamma\widetilde{\delta}\|_2^2 + \frac{1}{u_{j^*}}\left(\log\frac{1}{\delta_{j^*}} + \mathrm{KL}[\zeta:\pi]\right) + \xi \tag{63a}$$

$$\leq \frac{u_{j^*}}{8m}\|\Gamma\widetilde{\delta}\|_2^2 + \frac{3}{2u_{j^*}}\left(\log\frac{1}{\delta} + \mathrm{KL}[\zeta:\pi]\right) + \xi \tag{63b}$$

$$\leq \frac{1}{2}\|\Gamma\widetilde{\delta}\|_2\sqrt{\frac{\log\frac{1}{\delta} + \mathrm{KL}[\zeta:\pi]}{2m}} + \frac{3}{2}\|\Gamma\widetilde{\delta}\|_2\sqrt{\frac{\log\frac{1}{\delta} + \mathrm{KL}[\zeta:\pi]}{2m}} + \xi \tag{63c}$$

$$= 2\|\Gamma\widetilde{\delta}\|_2\sqrt{\frac{\log\frac{1}{\delta} + \mathrm{KL}[\zeta:\pi]}{2m}} + \xi \tag{63d}$$

Note that $\beta^*$ would attain the optimal value

$$\mathcal{R}(\zeta) - \mathcal{R}_m(\zeta, \mathcal{D}_m) \leq \|\Gamma\widetilde{\delta}\|_2\sqrt{\frac{\mathrm{KL}[\zeta:\pi] + \log\frac{1}{\delta}}{2m}} + \xi \tag{64}$$

which only differs from the above uniform bound by a factor of two. Finally, recall $\Gamma = \frac{\|\varrho\|}{d}D$ where $D$ has entries (17).

# B   Additional Lemmata

**Lemma 9.** *Let $T\colon \Omega \to \Omega$ be a measurable function on a measurable space $(\Omega, \Sigma)$ and let $\nu, \mu$ be measures on $\Omega$ with $T_\sharp\nu = \mu$. Let $B \in \Sigma$ be a fixed set with $\mu(B) > 0$ and $A = T^{-1}(B)$ its preimage under $T$. Then*

$$T_\sharp(\nu|A) = \mu|B\,. \tag{65}$$

*Proof.* Let $S \in \Sigma$ be arbitrary and let $\widetilde{\mu} = T_\sharp(\nu|A)$. Then

$$\widetilde{\mu}(S) = (\nu|A)(T^{-1}(S)) = \frac{\nu(T^{-1}(S) \cap A)}{\nu(A)} \tag{66}$$

as well as

$$(\mu|B)(S) = \frac{\mu(S \cap B)}{\mu(B)} = \frac{\nu(T^{-1}(S \cap B))}{\nu(A)} \tag{67}$$

Note that

$$z \in T^{-1}(S) \cap T^{-1}(B) \Leftrightarrow T(z) \in S \wedge T(z) \in B \Leftrightarrow T(z) \in S \cap B \Leftrightarrow z \in T^{-1}(S \cap B) \tag{68}$$

thus $T^{-1}(S) \cap T^{-1}(B) = T^{-1}(S \cap B)$ and consequently $\widetilde{\mu}(S) = (\mu|B)(S)$. Since $S$ was arbitrary, this shows the assertion. $\qquad\square$

The following theorem exists in various forms in the literature. To make this paper self-contained, we recite the version in [35] which is used to bound moment-generating functions in Proposition 4. Note that we only use the MGF bound (69) in our analysis. However, the concentration inequality (70) also holds analogously under the assumptions of Proposition 4 which may be of independent interest.

**Theorem 10 (Azuma-Hoeffding [35, Theorem 4.1]).** *Let* $(M^{(i)})_{i \in [m]}$ *be a martingale difference sequence with respect to a filtration* $(\Sigma_i)_{i \in [m]}$ *of sigma algebras. Suppose that for each* $i \in [m]$ *there exist* $\Sigma_{i-1}$*-measurable random variables* $A^{(i)}$, $B^{(i)}$ *such that* $A^{(i)} \leq M^{(i)} \leq B^{(i)}$ *almost surely. Then for all* $\lambda \in \mathbb{R}$ *it holds that*

$$\mathbb{E}\Big[\exp\big(\lambda \sum_{i \in [m]} M^{(i)}\big)\Big] \leq \exp\Big(\frac{\lambda^2}{8} \sum_{i \in [m]} \|B^{(i)} - A^{(i)}\|_\infty^2\Big) \tag{69}$$

*and consequently, for any* $t \geq 0$

$$\mathbb{P}\Big(\Big|\sum_{i \in [m]} M^{(i)}\Big| \geq t\Big) \leq 2\exp\Big(-\frac{2t^2}{\sum_{i \in [m]} \|B^{(i)} - A^{(i)}\|_\infty^2}\Big). \tag{70}$$

## C  Numerical Toy Example

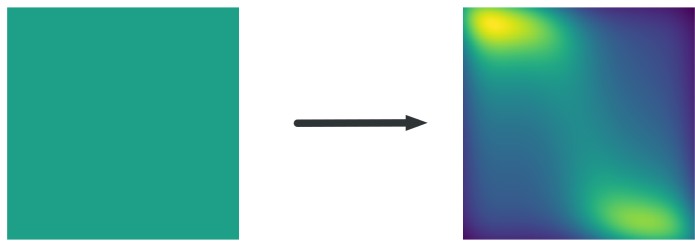

Figure 1: KR rearrangement $T(z) = (T_1(z_1, z_2), T_2(z_1, z_2))^\top$ transports a uniform reference measure $\nu^2$ on the unit cube $[0,1]^2$ to a multimodal distribution $\mu$ (right).

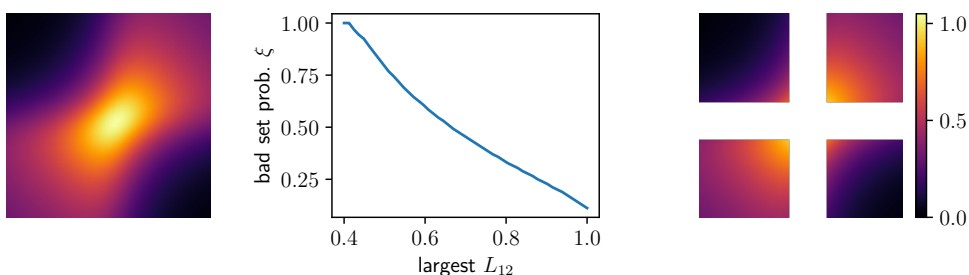

Figure 2: Construction of a good set in toy example. *Left*: $L_{12}$ for all inputs. *Center*: size of an excluded bad set under the data distribution for corresponding largest values of $L_{12}$. *Right*: $L_{12}$ for good inputs.

To illustrate the concept of transport stability discussed in Proposition 6, as well as the *bad set* construction of Section 5, we devised a toy example of two-dimensional transport. As a reference distribution, we choose the uniform distribution on the unit cube $[0,1]^2$, i.e. $\mathcal{Z} = [0,1]$. The metric $\rho^d$, $d = 2$ is chosen as the $\ell^1$-distance $\rho^2(z, z') = \|z - z'\|_1 = |z_1 - z_1'| + |z_2 - z_2'|$. The KR transport map has component functions $T(z_1, z_2) = \big(T_1(z_1), T_2(z_1, z_2)\big)$. We make a polynomial ansatz, defining

$$T_1(z_1) = \int_{[0,z_1]} \sum_{i \in [5]} \beta_i B_i^5(\tau) d\tau, \qquad T_2(z_1, z_2) = \int_{[0,z_2]} \sum_{i \in [2]} \widetilde{\beta}_i(z_1) B_i^2(\tau) d\tau \tag{71}$$

where $B_i^m$ denotes the $i$-th Bernstein polynomial of degree $m$, $\beta_i \geq 0$ are parameters and (positive) coefficient functions $\widetilde{\beta}_i(z_1)$ are again a parameterized combination of Bernstein polynomials (degree 8). Because Bernstein polynomials assume non-negative values on $[0, 1]$, the components in (71) define a valid KR-rearrangement. The resulting measure transport is illustrated in Figure 1. In order to evaluate the bound of Theorem 7, we need to compute the Lipschitz constant $L_{12}$ according to equation (16)

$$\mathbb{E}_{\tau \sim \nu_2}[|T_2(z_1, \tau) - T_2(z_1', \tau)|] \leq L_{12}|z_1 - z_1'| \tag{72}$$

for all good inputs $z_1, z_1'$. Figure 2 (left) shows the values of $L_{12}$ satisfying (72) for each input $(z_1, z_1') \in [0, 1]^2$. The sought Lipschitz constant is the largest of these values. In order to avoid regions with large values, Theorem 7 allows for the exclusion of *bad inputs*. The probability $\xi$ of this bad set under the data distribution is incurred as a penalty in equation (20). We thus construct bad sets to exclude large values and consider their probability under the data distribution Figure 2 (middle). Finally, the exclusion of such a bad set is shown on the right of Figure 2.

