# OpenReview forum: "On Certified Generalization in Structured Prediction"
_NeurIPS.cc/2023/Conference — NeurIPS 2023 poster_

### Official Review · Reviewer_H5Qp · 2023-06-19

**Soundness:** 3 good
**Presentation:** 2 fair
**Contribution:** 3 good
**Rating:** 6
**Confidence:** 3

**Summary:**

This paper proposes a PAC-Bayesian risk bound for the task of structured prediction. Under the assumption that the data is generated by  Knothe-Rosenblatt rearrangement, this method distills random output variables into a Wasserstein dependency matrix, which paves the way for improved generalization bounds of generative models.

**Strengths:**

1. The core method is based on the triangular measure transport and KR rearrangement of a tractable reference measure, which is novel and allows for flexible distillation.

2. The set of bad inputs is bounded so that the risk is more certificated.

3. The presented approach has the potential to compute risk certificates for many downstream discriminative tasks.

**Weaknesses:**

1. My main concern is that the samples are drawn from a distribution arising from a reference measure through KR rearrangement. Can authors do some toy experiments of some popular generative models to support this assumption?


**Questions:**

Please see the weaknesses and limitations

**Limitations:**

I had a rough check on the derivation and proofs, and they seem correct. My only concern is whether the assumption is valid in practice.

---

> ### Author Rebuttal · Authors · 2023-08-08
>
> # Thank you for your careful reading of our manuscript and insightful review
>
> It has been shown by [Bogachev2005] that any atom-free data distribution can be represented as the unique KR rearrangement of an atom-free reference distribution (such as the normal distribution or uniform distribution). This is a rather flexible model. For instance, every data distribution which has a density with respect to the Lebesgue measure satisfies these assumptions. As toy models, this includes all multivariate normal distributions and all Gaussian mixture models.
>
> Due to invertible architectures and measure transport required in both directions, these same assumptions are also at the core of many popular normalizing flow models, such as RealNVP [Dinh], FFJORD [Grathwohl], Invertible ResNet [Behrmann] and their conditional variants [Trippe], [Atanov].
> These models have been successfully used as surrogates for real data in multiple applications [Altekruger], [Kousha], [Lugmayr], [Horvat], [Wang] which empirically supports the assumption of an atom-free data distribution.
>
>
> [Dinh] Dinh, L., Sohl-Dickstein, J., & Bengio, S. (2016). Density estimation using real nvp. arXiv preprint arXiv:1605.08803.
>
> [Grathwohl] Grathwohl, W., Chen, R. T., Bettencourt, J., Sutskever, I., & Duvenaud, D. (2018). Ffjord: Free-form continuous dynamics for scalable reversible generative models. arXiv preprint arXiv:1810.01367.
>
> [Behrmann] Behrmann, J., Grathwohl, W., Chen, R. T., Duvenaud, D., & Jacobsen, J. H. (2019, May). Invertible residual networks. In International conference on machine learning (pp. 573-582). PMLR.
>
> [Trippe] Trippe, B. L., & Turner, R. E. (2018). Conditional density estimation with bayesian normalising flows. arXiv preprint arXiv:1802.04908.
>
> [Atanov] Atanov, A., Volokhova, A., Ashukha, A., Sosnovik, I., & Vetrov, D. (2019). Semi-conditional normalizing flows for semi-supervised learning. arXiv preprint arXiv:1905.00505.
>
> [Altekruger] Altekrüger, F., Denker, A., Hagemann, P., Hertrich, J., Maass, P., & Steidl, G. (2023). PatchNR: learning from very few images by patch normalizing flow regularization. Inverse Problems, 39(6), 064006.
>
> [Kousha] Kousha, S., Maleky, A., Brown, M. S., & Brubaker, M. A. (2022). Modeling srgb camera noise with normalizing flows. In Proceedings of the IEEE/CVF Conference on Computer Vision and Pattern Recognition (pp. 17463-17471).
>
> [Lugmayr] Lugmayr, A., Danelljan, M., Van Gool, L., & Timofte, R. (2020). Srflow: Learning the super-resolution space with normalizing flow. In Computer Vision–ECCV 2020: 16th European Conference, Glasgow, UK, August 23–28, 2020, Proceedings, Part V 16 (pp. 715-732). Springer International Publishing.
>
> [Horvat] Horvat, C., & Pfister, J. P. (2021). Denoising normalizing flow. Advances in Neural Information Processing Systems, 34, 9099-9111.
>
> [Wang] Wang, C., Zhu, Y., & Yuan, C. (2022, October). Diverse Image Inpainting with Normalizing Flow. In European Conference on Computer Vision (pp. 53-69). Cham: Springer Nature Switzerland.

---

> > ### Comment · Reviewer_H5Qp · 2023-08-13
> > **Thanks for thre response**
> >
> > Thanks for the response.
> >
> > The authors have provided a detailed explanation of  KR rearrangement. My concerns have been addressed. I will keep my original score of 6.

---

### Official Review · Reviewer_dT9a · 2023-06-27

**Soundness:** 3 good
**Presentation:** 3 good
**Contribution:** 3 good
**Rating:** 6
**Confidence:** 3

**Summary:**

This paper establishes a new PAC-Bayesian risk bound for the structured prediction problem. Technically, it assumes the data are generated by the Knothe-Rosenblatt rearrangement of a factorizing reference measure, and then obtains generalization bounds with the Wasserstein dependency matrix.

**Strengths:**

1. Different from the existing PAC-Bayesian bound with ϑ-mixing dependency matrix, this paper uses the Wasserstein dependency matrix to measure the interdependence of the data. Built upon the Wasserstein dependency matrix, novel concentration inequalities and general PAC-Bayesian risk bounds are established.

2. The Knothe-Rosenblatt rearrangement adopted in this paper is related to the measured transport and generative models, which may bring some insights into this field.


**Weaknesses:**

1. The discussion on the existing work [1*] should be clarified. The authors argue that [1*] requires that data are generated by a Markov random field. However, their general PAC-Bayesian risk bounds (Section 5 in [1*]) do not need this assumption. The ϑ-mixing dependency matrix adopted by them exists for any distribution. [1*] only choose Markov random fields as special examples to concretely clarify the implications of their theoretical framework. Thus, PAC-Bayesian risk bounds in this paper may not be more general than that in [1*].

2. There does not exist any theoretical example, simulation experiment, or real-world experiment in this paper. To show the superiority of the proposed framework, the authors should at least consider adding some concrete theoretical examples (e.g. Markov random field) in this paper.

3. The paper is not written very clearly. More descriptions and discussions can be added for the technical notions and definitions. For example, when I read this paper for the first time, some notions (e.g., Equation 7) confuse me, and I can not get intuitions of some definitions (e.g., Wasserstein dependency matrix).

I will consider improving the score after you discuss my concerns.

[1*] Ben London, Bert Huang, and Lise Getoor. Stability and generalization in structured prediction. JMLR 2016.


**Questions:**

Please see the weaknesses comments.

**Limitations:**

None.

---

> ### Author Rebuttal · Authors · 2023-08-08
>
> # Thank you for your careful reading of our manuscript and insightful review
>
>
> (1)
> We agree that, in their most abstract form, the results of [1*] do not require an MRF assumption. However, in our view, the key issue which reduces the relevance of this theory to practitioners is computability. This is the case especially since *tight* generalization bounds for deep learning via PAC-Bayesian theory have been more recently demonstrated [Dziugaite & Roy], [Perez], [Clerico] for finite sets of independent data. For example, [Clerico] certifies CIFAR-10 classification risk of at most 20.66% with probability at least 96.5% over the draw of the sample for a model with empirical test-set error of 19.52 +- 0.2 %. Thus, data dependency appears as the only remaining roadblock in structured prediction.
>
> With the more ambitious goal of tight, finite sample risk certificates in mind, the next step in our view is to instantiate abstract results of learning theory for concrete data models. Here, [1*] focuses on MRF data and all of the more concrete results in [1*] make this assumption. The authors explicitly leave the question of estimating ϑ-mixing coefficients from data for future work [1*, Section 7] and point to the preliminary work of [McDonald] on β-mixing. We are not aware of works which have pursued this for ϑ-mixing. In Section 2.1, we argue that this may not be possible without an explicit assumption on the data-generating process.
>
> This is the starting point of our current work, we choose a specific, very general data model (measure transport by KR rearrangement) and construct an analogous PAC-Bayesian bound from related concentration of measure results which play particularly nicely with this data model.
> More specifically, the key technical differences between our mathematical framework and theirs are (a) the choice of coupling measure between conditional distributions and (b) the choice of metric on the data space. They use the coupling measure construction of [Fiebig] and the Hamming distance which allows to distill dependency into a ϑ-mixing dependency matrix [1*, Lemma 6 in Appendix A.2]. We construct the required coupling from properties of the KR rearrangement (Lemma 5 and Lemma 3) and apply the concentration of measure results of [Kontorovich] without choosing a specific metric. Our approach is thus specifically geared towards measure transport models by essentially translating data dependency into a property of the transport map (the Lipschitz constants $L_{ij}$ in Eq. (16), Proposition 6).
>
> (2)
> We evaluated eq. (16) of Proposition 6 for a 2D-toy scenario and defined the bad set as suggested in the first paragraph of Section 5, line 297, in order to demonstrate that the theory is amenable to numerical evaluations in principle. Please see the PDF attached for an illustration. As mentioned in the paper, the development of dedicated numerical methods and experiments is beyond the scope of the paper, however.
>
> (3)
> Regarding eq. (7), we will add the following explanatory text to line 152:
> Here $K^{(i)}(x,dy)$ is a Borel measure for every $x$ and (8) computes the expected value of $f$ at x, conditioned on the fixed realization of the subvector $x^{[i-1]}$.
>
> The subsequent sentence in the paper, preceding Definition 2, clarifies the role of the Wasserstein matrix: "It turns out that the effect of the kernel (7) on local oscillations serves to quantify dependence of data with joint distribution $\mu$."
>
>
> [1*] London, B., Huang, B., & Getoor, L. (2016). Stability and generalization in structured prediction. The Journal of Machine Learning Research, 17(1), 7808-7859.
>
> [Dziugaite & Roy] Dziugaite, G. K., & Roy, D. M. (2018). Data-dependent PAC-Bayes priors via differential privacy. Advances in neural information processing systems, 31.
>
> [Perez] Pérez-Ortiz, M., Rivasplata, O., Shawe-Taylor, J., & Szepesvári, C. (2021). Tighter risk certificates for neural networks. The Journal of Machine Learning Research, 22(1), 10326-10365.
>
> [Clerico] Clerico, E., Deligiannidis, G., & Doucet, A. (2022, May). Conditionally gaussian pac-bayes. In International Conference on Artificial Intelligence and Statistics (pp. 2311-2329). PMLR.
>
> [McDonald] Mcdonald, D., Shalizi, C., & Schervish, M. (2011, June). Estimating beta-mixing coefficients. In Proceedings of the Fourteenth International Conference on Artificial Intelligence and Statistics (pp. 516-524). JMLR Workshop and Conference Proceedings.
>
> [Fiebig] Fiebig, D. (1993). Mixing properties of a class of Bernoulli-processes. Transactions of the American Mathematical Society, 338(1), 479-493.
>
> [Kontorovich] Kontorovich, A., & Raginsky, M. (2017). Concentration of measure without independence: a unified approach via the martingale method. In Convexity and Concentration (pp. 183-210). Springer New York.

---

> > ### Comment · Reviewer_dT9a · 2023-08-12
> >
> > Thanks very much for the detailed response. I appreciate that you clarify the discussion on the existing work [1*] and add a simulation to support your theory. I have increased the score from 4 to 6.
> >
> > [1*] Ben London, Bert Huang, and Lise Getoor. Stability and generalization in structured prediction. JMLR 2016.

---

### Official Review · Reviewer_uxkb · 2023-06-27

**Soundness:** 3 good
**Presentation:** 3 good
**Contribution:** 3 good
**Rating:** 6
**Confidence:** 2

**Summary:**

This work derives a novel PAC-Baeysian risk bound for structured prediction based on generative models, a triangular and monotone transport and Wasserstein dependency matrices.

**Strengths:**

This is technical a paper with rigorous theoretical analysis. The flow is easy to follow. The authors have made very detailed comparisons to [1], on which this work is largely based. Many examples such as image data are included to help readers understand some of the intuitions behind the derived theoretical results.


[1] London, Ben, Bert Huang, and Lise Getoor. "Stability and generalization in structured prediction." The Journal of Machine Learning Research 17, no. 1 (2016): 7808-7859.

**Weaknesses:**

There are no empirical results to showcase the tightness or usefulness of the risk bounds compared to bounds that ignore the size of the structure object. The assumptions are probably limited. See my comments on these points below.

**Questions:**

I have a few minor concerns as follows:

1. In line 134, it is assumed that the structured loss is an additive, bounded and pointwise loss. This looks like the conclusion in this paper can only be generalized to affine losses. This assumption does not hold for, e.g., log loss, which goes to infinity, or F1 score, which is non-decomposable into components. So what family of losses can the risk bound generalize to?

2. The successful derivation of the risk bound in Theorem 7 is greatly thanks to, I believe, the introduction of the bad set. I appreciate the detailed explanations on the intuitions for this set. But in practice, how do you define bad and how can you identify those data points? As the authors suggest by themselves, one could use Equation 16 to decide on this but the Lipschitz constants $L_{ij}$ may not be easy to compute.

3. Does the learned measure transport distribution converge to the true underlying distribution asymptotically? If not, how close are they?


**Limitations:**

Yes, the authors have adequately discussed the limitations of their findings.

---

> ### Author Rebuttal · Authors · 2023-08-08
>
> # Thank you for your careful reading of our manuscript and insightful review
>
> Bounds which ignore the size of the structured object can not make any useful statement on generalization from a single example. For instance, the typical setting of node classification with graph neural networks considers a single graph, with training labels available on a subset of the nodes. In this case, all available data is part of a single structured object (the graph) and PAC-Bayesian learning theory (the only theory which is currently able to achieve tight bounds in deep learning) can not make a useful statement about generalization if the internal structure is ignored (m = 1).
>
>
> # Response to Reviewer Questions
>
> (1)
> The assumptions made on the loss function are common in the PAC-Bayesian literature.  Recent works have aimed to relax them with some success. We refer to these works [3,27,28] in Section 1.1, line 56, an even more recent example is [Haddouche]. However, generalization to unbounded losses requires nontrivial extensions to our proofs.
> Decomposition of the loss into a sum of component losses is usually assumed in PAC-Bayesian constructions as a prerequisite to underlying concentration of measure results (such as Hoeffding's inequality, which makes a statement about the sum of independent variables). Since we use more general concentration of measure theory which handles dependent data, our results can more easily generalize to losses which do not decompose (but are still bounded).
>
> Note that PAC-Bayesian theory also offers opportunities. For example, the construction does not require differentiable loss. Thus, some loss functions (such as the 01 loss which is particularly natural in classification) can actually be available for PAC-Bayesian risk certification even though they are not typically useful for the training of deep networks. An example is the model of [Clerico] which enables direct optimization of expected generalization error in the sense of 01 loss. Beyond 01 loss, it has been proposed to certify the confusion matrix of classifiers using PAC-Bayesian methods [Adams].
>
>
> (2)
> We evaluated eq. (16) of Proposition 6 for a 2D-toy scenario and defined the bad set as suggested in the first paragraph of Section 5, line 297, in order to demonstrate that the theory is amenable to numerical evaluations in principle. Please see the PDF attached for an illustration. As mentioned in the paper, the development of dedicated numerical methods and experiments is beyond the scope of the paper, however.
>
> (3)
> Under the assumptions adopted in the paper, the measure transport can be realized via KR rearrangement, in principle; see, e.g., [6] (mentioned in line 90). Convergence rates for finite sample regimes, however, define an open research problem. A step in this direction was recently made in [4]. This is mentioned in the "Limitations" Section, line 341.
>
> [Haddouche] Haddouche, M., & Guedj, B. (2023). Wasserstein PAC-Bayes Learning: A Bridge Between Generalisation and Optimisation. arXiv preprint arXiv:2304.07048.
>
> [Clerico] Clerico, E., Deligiannidis, G., & Doucet, A. (2022, May). Conditionally gaussian pac-bayes. In International Conference on Artificial Intelligence and Statistics (pp. 2311-2329). PMLR.
>
> [Adams] Adams, R., Shawe-Taylor, J., & Guedj, B. (2022). Controlling confusion via generalisation bounds. arXiv preprint arXiv:2202.05560.

---

> > ### Comment · Reviewer_uxkb · 2023-08-20
> >
> > Thanks for your rebuttal. I have also read the other reviews. I will keep my rating and lean towards acceptance.

---

### Official Review · Reviewer_kVX1 · 2023-07-06

**Soundness:** 2 fair
**Presentation:** 3 good
**Contribution:** 3 good
**Rating:** 6
**Confidence:** 4

**Summary:**

This paper develops a PAC-Bayesian bound on the risk of structured predictors which decreases with the number and size of examples. The work builds on concentration of measure results (e.g., [33]) and continues the line of work in [36] by removing the assumption that data are generated by a Markov Random Field (MRF). Instead, the present work assumes a triangular and monotone transport, a Knothe-Rosenblatt (KR) rearrangement of a reference measure, as the data model.

**Strengths:**

**S1.** This theoretical work advances a series of works on bounding the risk of structured predictors via more recent concentration of measure results.

**S2.** The submission does a great job at providing an overview of prior results and presenting new developments.

**Weaknesses:**

**W1.** The theoretical results in this submission are presented with little connection to implications to practice. For example, L341-344 state that how closely a measure transport distribution learned from data approximates the actual (unknown) distribution of the data is an open question and thus, no empirical results are provided. I wonder if more could be said in this regard.

As another example, in [36] bounds are applied to specific models. On one hand, the specifics of the model and training loss are taken into account. On the other hand, the PAC-Bayes bound is derandomized in order to apply it to a learned predictor. Are similar applications not possible for the results in the present submission?

**W2.** One of the claims is that the present work “makes a preliminary step towards leveraging powerful generative models to establish generalization bounds for discriminative downstream tasks.” In my view, providing further insight into next steps in this direction would help other researchers build upon this work.

**Questions:**

**Q1.** How are the theoretical results in this submission (perhaps given further development) to inform practice?

**Q2.** In general, what would be possible progressions of developments building on this work?

**Q3.** How does the restriction to atom-free data distributions limit relevance and applicability of the results in this submission, e.g., to discrete output spaces?


**Limitations:**

Potential negative societal impact was not discussed.

---

> ### Author Rebuttal · Authors · 2023-08-08
>
> # Thank you for your careful reading of our manuscript and insightful review
>
> # W1 & Q2
> The recent work of [Baptista] discusses a parametric class of monotone triangular maps which can serve as approximations of KR rearrangement. A possible direction of future work is to specialize this construction to functions which in addition provide a tractable form of the Lipschitz constants in Eq. (16).
>
> The derandomization proposed in [London, Proposition 4] is built on an assumption on hypothesis stability. This is orthogonal to the assumptions on input stability made in our work and thus applies analogously in concert with our PAC-Bayesian construction. We have omitted this step as well as the consideration of bad hypotheses proposed by [London] for ease of exposition (see lines 289-290). We will add reference to this after line 290 "The same applies to the derandomization strategy proposed by [London] which is based on hypothesis stability."
>
>
> # W2 & Q1
> Our work connects state of the art PAC-Bayesian learning theory (the only theory which is currently able to achieve tight bounds in deep learning) to plausible assumptions about concrete generative data models used in practice and realized via KR-based measure transport. In particular, our work addresses the challenging case of structured prediction and the ability to learn from few samples due to the dependency caused by internal structure.
>
> A specific use of this theory is to improve training procedures based on a better understanding of generalization. Because tight risk certificates are now available in PAC-Bayesian learning, it has become possible to directly optimize a (tight) bound on the generalization error (out-of-sample) rather than merely optimizing empirical risk (in-sample) [Clerico] [Perez]. Our work is towards extending this to structured prediction scenarios. As an example, node-classification with graph neural networks can not currently be studied with established PAC-Bayesian methods because, although labels are available on a subset of graph nodes, all data are dependent (m = 1).
>
> # Q3.
> Due to invertible architectures and measure transport in both directions, the assumption of atom-free data distributions is required by the theory of measure transport based on KR rearrangements and hence at the core of corresponding popular normalizing flow models used today (such as RealNVP [Dinh], FFJORD [Grathwohl], Invertible ResNet [Behrmann]). This excludes discrete output spaces. On the other hand, due to the well-established theory of quantization of probability distributions, discrete scenarios can be implicitly modeled using atom-free measures, too - see, e.g., the scenario mentioned in the lines 127-128.
>
> [London] London, B., Huang, B., & Getoor, L. (2016). Stability and generalization in structured prediction. The Journal of Machine Learning Research, 17(1), 7808-7859.
>
> [Baptista] Baptista, R., Marzouk, Y., & Zahm, O. (2020). On the representation and learning of monotone triangular transport maps. arXiv preprint arXiv:2009.10303.
>
> [Clerico] Clerico, E., Deligiannidis, G., & Doucet, A. (2022, May). Conditionally gaussian pac-bayes. In International Conference on Artificial Intelligence and Statistics (pp. 2311-2329). PMLR.
>
> [Perez] Pérez-Ortiz, M., Rivasplata, O., Shawe-Taylor, J., & Szepesvári, C. (2021). Tighter risk certificates for neural networks. The Journal of Machine Learning Research, 22(1), 10326-10365.
>
> [Dinh] Dinh, L., Sohl-Dickstein, J., & Bengio, S. (2016). Density estimation using real nvp. arXiv preprint arXiv:1605.08803.
>
> [Grathwohl] Grathwohl, W., Chen, R. T., Bettencourt, J., Sutskever, I., & Duvenaud, D. (2018). Ffjord: Free-form continuous dynamics for scalable reversible generative models. arXiv preprint arXiv:1810.01367.
>
> [Behrmann] Behrmann, J., Grathwohl, W., Chen, R. T., Duvenaud, D., & Jacobsen, J. H. (2019, May). Invertible residual networks. In International conference on machine learning (pp. 573-582). PMLR.

---

> > ### Comment · Reviewer_kVX1 · 2023-08-17
> >
> > Thank you for the response. In my view, even though the contribution is theoretical, commentary that bridges theory to practical implication or future development will benefit the submission -- e.g., the possibility of directly optimizing generalization error.
> >
> > After reading other reviews and author responses I am keeping my original score.

---

### Official Review · Reviewer_zkw9 · 2023-07-25

**Soundness:** 4 excellent
**Presentation:** 3 good
**Contribution:** 2 fair
**Rating:** 6
**Confidence:** 3

**Summary:**

This paper studies PAC-Bayesian risk bound for structured prediction using concentration of measure. Specifically, authors characterize stability and dependency with Wasserstein dependency matrix under the data assumption that data are given by Knothe-Rosenblatt (KR) rearrangement of a factorizing reference measure. PAC-Bayesian risk bound for structured prediction has been studied in the previous studies [1], [2] with different stability notions and the main contribution of this paper is to apply measure-theoretic stability notions from [3] to the problem and provide a new PAC Bayesian bound that scales with the number of examples and their dimensions. This result agrees with the PAC risk bound provided in [2] under specific conditions, giving a more generalized interpretation based on Wasserstein dependency matrix.

[1] London, Ben, et al. "Collective stability in structured prediction: Generalization from one example." International Conference on Machine Learning. PMLR, 2013.

[2] London, Ben, Bert Huang, and Lise Getoor. "Stability and generalization in structured prediction." The Journal of Machine Learning Research 17.1 (2016): 7808-7859.

[3] Kontorovich, Aryeh, and Maxim Raginsky. "Concentration of measure without independence: a unified approach via the martingale method." Convexity and Concentration. Springer New York, 2017.

**Strengths:**

1. New stability notion is considered for PAC bayes risk bound, providing additional analysis and interpretations up on previous works.
2. Provided examples and intuitions are helpful for readers to follow the manuscript.
3. KR rearrangement assumption on data is not that unrealistic, considering that generative models using KR rearrangement assumption have been studies recently, e.g. [4]. This entails the possibility of practical implications.


[4] Irons, Nicholas J., et al. "Triangular flows for generative modeling: Statistical consistency, smoothness classes, and fast rates." International Conference on Artificial Intelligence and Statistics. PMLR, 2022.

**Weaknesses:**

1. Definitions, notations, some lemma & theorem look pretty similar to ones in [3] --- authors may want to highlight the contribution of this paper by differentiating from [3] or giving more explanation.
2. More background on PAC Bayes risk bounds could be helpful for readers. Especially, it would be nicer if how stability and dependency affects PAC Bayes risk bounds can be discussed.

**Questions:**

1) If measure space is discrete, can it be reduced to previously known PAC Bayes bounds? I am curious how measure-theoretic characterization can be connected with the previous results.
2) What's the intuition about "a bad set" --- is it the set possibly locally unstable?
3) Is there any possible simulation setup we can see the provided PAC Bayes bound works?
4) L151: What is $\delta_x$? --- confusion with $\delta_i$. $\delta$ is also overloaded in the probability $1-\delta$, authors may want to change the notation to reduce confusion.


**Limitations:**

Limitations are properly discussed.

---

> ### Author Rebuttal · Authors · 2023-08-08
>
> # Thank you for your careful reading of our manuscript and insightful review
>
> 1.
> We do not claim to make a contribution to the mathematical literature on measure concentration with respect to functions of dependent random variables. Rather, we contribute to machine learning by applying the concepts devised in [3] to *structured* prediction based on the class of generative data models (normalizing flows) based on KR rearrangement. Thus our work extends the recent line of research on tight PAC bounds for deep learning to structured prediction, based on a probabilistic data model which covers many practical scenarios.
>
> 2.
> We refer in Section 1.1, line 43 to works which survey and introduce to PAC-Bayes theory. The influence of stability and dependency on our novel PAC-bound is discussed in the paragraph directly after the main Theorem 7, lines 245-250: bounded dependency and large size d of data points can compensate for smaller numbers m of samples in order to achieve tight bounds, which is important for scenarios of *structured* predicion.
>
>
> # Response to Reviewer Questions
>
> 1.
> Since our approach relies on a generative data model (normalizing flows) via KR rearrangement, which comprises an invertible architecture and measure transport in both directions (training and sampling), we assume that both the reference and the target (data) distribution are atom-free, according to the mathematical theory underlying KR-based measure transport. This excludes discrete spaces. On the other hand, due to the well-established theory of quantization of probability distributions, discrete scenarios can be implicitly modeled using atom-free measures, too - see, e.g., the scenario mentioned in the lines 127-128.
>
> The format of our novel PAC bound, eq. (20), clearly connects to prior known PAC bounds. The right-hand side has the ususal format comprising the empirical risk and a complexity term. The essential difference is that the latter term depends also on the norm of the Wasserstein dependency matrix times the oscillation vector, which enables to learn and reliably predict even from few examples (small m) provided the internal dependencies are bounded and the size d of each data point is large.
>
> 2.
> The intuition about the bad set are rare data points whose internal dependency and structure, which we explore to learn and certify structured prediction from few examples, does *not* conform to the quantitative assessment of this internal dependencies, according to Proposition 6. The bad set contains data points with `unusually' pronounced internal dependencies, causing instability under data perturbations. Thus, the concept of a bad set enables to quantify expected internal data dependencies in a sensible way (Proposition 6) so as to make tight the novel PAC-Bayes risk bound for structured prediction (Theorem 7), for any typical (= good) data point.
>
> 3.
> We evaluated eq. (16) or Proposition 6 for a 2D-toy scenario and defined the bad set as suggested in the first paragraph of Section 5, line 297, in order to demonstrate that the theory is amenable to numerical evaluations. Please see the PDF attached for an illustration. As mentioned in the paper, the development of dedicated numerical methods and experiments is beyond the scope of the paper.
>
> 4.
> We agree and are aware that the symbol $\delta$ is overloaded, which is acceptable if both the argument and sub- or superscripts disambiguate the interpretation. In the present case, $\delta_{x}$ with a vector as subscript denotes a Dirac measure in order to make Markov kernels well-defined in connection with conditional probability laws (eq. (7)) and expectation (eq. (8)). On the other hand, $\delta_{i}$ with a number in $[d]$ as subscript and a function as argument (e.g. eqns. (6) and (9)) uniquely refer to the local oscillation of the argument.

---

> > ### Comment · Reviewer_zkw9 · 2023-08-11
> >
> > I've read the authors' rebuttal and other reviews. The authors clarified main contributions and nicely addressed my questions. Still, as this paper applies the concepts devised in [3], it could have included more descriptions and explanations about them with more context. Thus I would keep my score --- I am inclined to accept based on its contribution, but I am also not strongly against rejecting the paper.

---

### Author Rebuttal · Authors · 2023-08-08


# We thank all reviewers for their insightful and constructive reviews.

We respond in detail to each reviewer in the corresponding sections. Below, we summarize our responses to points which were raised by at least two reviewers.

# Connections/implications to practice, KR-based generative models and real data distributions.
Our work connects the recent line of research on tight PAC bounds for deep learning to *structured* prediction, based no generative models via KR-arrangements. The latter can cover any atom-free real data distribution in theory and a broad range of realistic scenarios in practice (our detailed response provides more references). Approximation bounds for finite sample scenarios is the subject of ongoing research, see, e.g., [4] for a step towards this goal.

# Discrete spaces
KR-based generative models require measure transport in both directions and assume atom-free reference and target distributions; see, e.g., [11]. We do not consider this as a serious restriction since discrete scenarios can be represented through the quantization of probability distributions.

# Toy experiments, intuition about bad sets
We evaluated a toy experiment to demonstrate that our approach is amenable to numerical evaluation, in principle. Please see the PDF sheet attached. The design of a dedicated numerical algorithm and experiments is beyond the scope of the paper, however.

---

### Decision · Program_Chairs · 2023-09-21

**Decision:**

Accept (poster)

**Comment:**

This paper tackles learning theory problems in structured prediction, a more challenging setting for theoretical results compared to standard binary or multiclass classification. The goal is to provide generalization bounds that incorporate the richness of the structures of the samples. In particular, in such problems there is a tradeoff between the number of independent samples and the "size" of each sample, where component parts are not necessarily independent.

The authors effectively generalize existing results in this area by relying on a weaker assumption on the data generating process. The generalization is reasonably strong (moving from requiring Markov random fields to a much more general class of distributions). The authors overcome a number of technical hurdles in order to be able to do this. This culminates in a nice PAC-Bayes result, the central result of the paper.

While the result is purely theoretical, it will be appreciated by researchers in structured prediction and is likely to be a useful building block for future learning theory results in the area. As the authors note, it is also a good step towards building practical results for downstream use.

For these reasons, I recommend acceptance. Additionally, all reviewers also leam accept and had their questions resolved by the authors during the rebuttal stage.